# Kinetic Modeling of Grain Boundary Diffusion: The Influence of Grain Size and Surface Processes

**DOI:** 10.3390/ma13051051

**Published:** 2020-02-26

**Authors:** Justina Jaseliunaite, Arvaidas Galdikas

**Affiliations:** Physics Department, Kaunas University of Technology, Studentu st., LT-51368 Kaunas, Lithuania; justina.jaseliunaite@ktu.edu

**Keywords:** polycrystals, mass transfer, grain boundary diffusion, adsorption, kinetic modeling, rate equations, solid oxide fuel cells

## Abstract

Based on rate equations, the kinetics of atom adsorption, desorption, and diffusion in polycrystalline materials is analyzed in order to understand the influence of grain boundaries and grain size. The boundary conditions of the proposed model correspond with the real situation in the electrolytes of solid oxide hydrogen fuel cells (SOFC). The role of the ratio of grain boundary and grain diffusion coefficients in perpendicular and parallel (to the surface) concentration profiles is investigated. In order to show the influence of absolute values of grain and grain boundary diffusion coefficients, we select four different cases in which one of the diffusion coefficients is kept constant while the others vary. The influence of grain size on diffusion processes is investigated using different geometrical models. The impact of kinetic processes taking place on the surface is analyzed by comparing results obtained assuming the first layer as a constant source and then involving in the model the processes of adsorption and desorption. It is shown that surface processes have a significant influence on the depth distribution of diffusing atoms and cannot be ignored. The analytical function of overall concentration dependence on grain and grain boundary volume ratio (*V_g_/V_gb_*) is found. The solution suggests that the concentration increases as a complementary error function while *V_g_/V_gb_* decreases.

## 1. Introduction

Grain boundary (GB) diffusion plays an important role in the mass transport process in polycrystalline materials. In most cases, simplified models are used to model the behavior of GB diffusion. Idealized geometries were first developed by Fisher, where the GB is assumed as a separate medium of width δ inserted in bulk perpendicularly to the free surface and the concentration change across it is negligible [1]. The diffusion coefficient remains constant along the GB (is isotropic) and is independent of concentration. The GB diffusion coefficient is much higher than the diffusion coefficient in bulk. The Fisher model is still important in GB diffusion theory, but it has been extended and modified. A cubic grain model with instantaneous and constant source was considered by Suzuoka [2] and Whipple [3,4]. The GB region was isolated and was isotropic with high diffusivity. Another study was done based on the Fisher model whereby the diffusivity in micro- and nano-crystalline structures was analyzed [5]. It was shown that diffusion is faster in nanograin boundaries than in micrograin boundaries, and faster in nanograins than in micrograins. The activation energy needed for the processes is similar [5], so the size of the polycrystalline material grains influences the mass transport process. Mishin used the same Fisher model, but considered the anisotropy and the spatial inhomogeneity of the GB diffusion coefficient [6,7].

Harrison’s A-B-C classification describes the kinetic regimes of diffusion in a polycrystal with parallel GB and is also based on Fisher’s model [7]. Regimes differ by diffusion parameters. In regime A, the diffusion length is larger than the spacing between GB. In regime B, grain boundary diffusion takes place with bulk diffusion from the boundary into the grain. However, the grain boundaries are distributed farther apart than in the A regime and can be assumed to be isolated. In the C regime, diffusion takes place only along GBs without any or with insignificant leakage to grain [1,7]. Le Claire developed a coefficient called the Le Claire parameter β, which describes the scale at which diffusion within crystallites is relatively stronger than volumetric diffusion. At bigger β values the concentration contours are more likely to be curved along grain boundaries, which means that more leakage from grain boundaries to grains occurs [1].

To make diffusion happen, there should be adsorption of gas molecules on the surface. According to J.H. de Boer, gas can be imagined as a huge number of molecules that travel in all directions, collide with each other, and can approach the surface and hit it [8]. Then there are two options: they can bounce off, or adsorb on the surface. The latter option is more often occurrent, but afterwards an atom can be desorbed or diffused into the material volume [9]. The Langmuir adsorption model explains how adsorbates behave in ideal isothermal conditions [10,11]. There are several assumptions in this Langmuir model: the surface on which the substance adsorbs is completely flat and smooth, adsorption takes place only at certain adsorption centers, they are evenly distributed on the adsorbent surface, and only one atom can adsorb in one center. At that center, the adsorbed atom can desorb, and the other atom may adsorb in the newly emerging center. Adsorbed atoms do not interact with each other and all adsorption centers have the same energy.

To gain more detailed knowledge about GB diffusion, different investigation and calculation methods were used. Gryaznov, Fleig, and Maier [12] used numerical finite element simulation with the modified Fisher method. In the study, they investigated GB both parallel and perpendicular to the surface. They concluded that if the GB diffusion length is larger than the grain size, GB perpendicular to stream source has a greater influence on the mass transport process. The studies were performed at different Harrison’s diffusion regimes, when grains are square and their size differs [13,14]. In [15], two-dimensional grain patterns were constructed, where first the grains are arranged one after another and then they are distributed as brickwork. These patterns were used to determine the effective diffusivity using the Hart-Mortlock and Maxwell-Garnett equations in a Monte Carlo simulation. They provided a good determination of the effective diffusivity by changing the ratio of the diffusion coefficient in grain and GB.

Some research uses not cubic grains, but more complex geometry—Voronoi grain distribution [16,17,18]. The Voronoi model divides a region into polygons that fill the space without any overlap. Others compare single uniform GB and a complex boundary network [19]. With numerical and analytical methods, they investigated impact of complex microstructure, which is more realistic. They found out that the concentration distribution depends on the grain boundary geometry and on the relationship between the grain boundary diffusion and grain diffusion coefficients [16]. The bigger the difference, the faster the mass transport. Likewise, if coefficients differ much then the influence of GB is increasing, because atoms penetrate through GB and leak into the grain. Investigations were done comparing polycrystal and bicrystal materials in order to get information about segregation and the effect of moving boundaries [20]. Also, studies were made in order to evaluate the impact of GB activation energy variability in the mass transport process [17]. Different diffusion regimes can be identified by the activation energy. Grain boundaries can act not only as fast diffusion paths but as a sink, which slows down diffusivity, because atoms can be trapped in grain boundaries [21]. Experiments were performed whereby the influence of grain size [22], activation energies [23], and grain boundary energies [24] on mass transportation were investigated.

One case where GB diffusion is very important is solid state electrolytes or superionic conductors [25]. Depending on the superionic granular structure, the crystalline and intercrystallite ionic conductivity changes. Some researchers found that a decrease in grain size guarantees better ionic conductivity [26]. However, this statement is only valid when the superionic conductor is a nanocrystalline material because grain boundaries can decrease the ion transportation process while increasing impermeability [27]. Investigations were done to ensure that decreasing the grain size to nanometers will increase ion conductivity [28,29,30]. These proved that coarser grains are more distributed when higher ionic conductivity is achieved.

In this work we want to obtain a better understanding of how atoms are transferred through different types of geometry of nanocrystalline materials, to ascertain how grain boundaries and their occupied area influence the concentration change in grains. Also, we hope to show what the influence of processes on the surface is, especially adsorption and desorption, because most of the models ignore this question and use a constant source in the first layer. The purpose of this research is to develop a tool (model and code) to consider the dynamics and mechanisms of the diffusion of atoms and ions in polycrystalline materials, of which many aspects are not fully understood, especially for ions in superionic materials. The boundary conditions of the proposed model correspond with the real situation of electrolytes of solid oxide hydrogen fuel cells (SOFC).

## 2. Kinetic Model

The presented model is based on Fick’s second law and the Langmuir adsorption equation. We consider the process of grain boundary diffusion in terms of the random walk of particles in a polycrystalline material. For a two-dimensional case when the diffusion coefficient is constant, Fick’s second law is written as follows:(1)$$\frac{\partial c\left(x,y,t\right)}{\partial t}=D_{x}\frac{\partial^{2}c\left(x,y,t\right)}{\partial x^{2}}+D_{y}\frac{\partial^{2}c\left(x,y,t\right)}{\partial y^{2}}$$

Assuming that the diffusion coefficient is independent of direction *D_x_ = D_y_ = D* (this assumption is widely used for isotropic materials [16,17,19]), and the atomic layer thickness is the same for all directions *h_x_ = h_y_ = h*, Equation (1) can be rewritten in numerical form as [31]:(2)$$c_{t+\Delta t}^{i,j}=c_{t}^{i,j}+\left(\frac{D}{\;h^{2}}\left(c_{t}^{i+1,j}+c_{t}^{i-1,j}-2c_{t}^{i,j}\right)+\frac{D}{\;h^{2}}\left(c_{t}^{i,j+1}+c_{t}^{i.j-1}-2c_{t}^{i,j}\right)\right)\Delta t$$
where Δt  is the time step, *i* describes a vertical coordinate, and *j* describes a horizontal coordinate. The same equation is used for both grain and GB diffusion, but the difference is in the value of diffusion coefficient *D*: we use *D = D_gb_* for diffusion in GB and *D = D_g_* for diffusion in grains. Also, the same equation is used for atom transfer from grains to grain boundaries and vice versa: when atoms diffuse from GB to grain, then *D = D_g_*; when from grain to GB, *D = D_gb_*.

Many reports consider the first layer with constant concentration, ignoring adsorption and desorption processes on the surface. In this model, the processes of adsorption and desorption are included. According to the Langmuir adsorption model, the rate of adsorption is proportional to the gas pressure and number of adsorption centers. The rate of desorption is proportional to the number of adsorbed atoms [10]. The Langmuir equation is as follows:(3)$$\frac{dc}{dt}=\alpha p\left(c^{*}-c\right)-\beta c$$
where α is the adsorption coefficient, β is the desorption coefficient, *c* and *c** are the concentration of adsorbate and concentration of adsorption centers, respectively, and *p* is the gas pressure. In the model the first layer *i = 1* is the surface, where adsorption and desorption take place. Including the Langmuir equation in Equation (2), the equation for the first layer becomes the following [32,33]:(4)$$c_{t+\Delta t}^{1,j}=c_{t}^{1,j}+\left(\alpha p\left(c^{*}-c_{t}^{1,j}\right)-\beta c_{t}^{1,j}-\frac{D}{\;h^{2}}\left(c_{t}^{1,j}-c_{t}^{2,j}\right)+\frac{D}{\;h^{2}}\left(c_{t}^{1,j+1}+c_{t}^{1.j-1}-2c_{t}^{1,j}\right)\right)\Delta t$$

This model assumes: (1) Surface adsorption and desorption; (2) Volumetric diffusion from surface to deeper layers; (3) Diffusion from grain to GB; (4) Diffusion from GB to grain; (5) Diffusion along and across the GB; and (6) Desorption of the last layer (both grain and GB). As an initial condition, we selected zero concentration of diffusant atoms in the whole volume. The solver was developed using standard C++ libraries, based on the finite differences method and an explicit discretization scheme. A schematic presentation of the solver is shown in Figure 1. Boundary conditions are as follows: atoms adsorb onto the surface; some of them can desorb while another part penetrates the surface layer and diffuses into the volume of the material according to 2D geometry. Those atoms that reach the bottom edge layer may desorb. Desorption from the lateral edge layers is excluded. Such boundary conditions allow us to simulate the mass transfer processes in polycrystalline electrolytes of solid oxide hydrogen fuel cells (SOFC), where oxygen ions diffuse from a cathode to an anode that is placed on another side of the electrolyte layer and then desorb after recombination with hydrogen. Removal (and arrival) of particles from (to) the lateral surfaces is excluded or negligible. All the calculated results presented below are outputted at a certain arbitrary time.

In this work, five geometrical models with different grain sizes are considered (see Figure 2). Grain size varies: 30 a.u. model (a), 42 a.u. model (b), 66 a.u. model (c), 90 a.u. model (d) and 138 a.u. model (e). GB size is fixed (6 a.u.) in all models. Table 1 shows a numerical comparison of the two-dimensional volumes of each model. Most of the total area occupied by GB is in model (a), with the least in model (e).

Many reports consider models with square grains surrounded by GB [12,15,19,34]. Very often, they consider that the concentration within GB width is constant and unchanging. In this work a change of concentration within GB during the diffusion process is allowed.

## 3. Results and Discussion

The most important physical parameter considering the influence of grain boundary diffusion is the relative diffusion coefficient *D_gb_/D_g_*, the ratio of GB diffusion coefficient *D_gb_* to grain diffusion coefficient *D_g_*. In order to consider the impact of individual diffusion coefficient *D_gb_* and *D_g_* the calculations were performed for four different cases (see Table 2). Ratio *D_gb_/D_g_* in the interval 10^2^–10^4^ is most usually taken for consideration [19]. Two different values of relative diffusion coefficient are considered, 10^3^ and 10^4^, when *D_gb_* is fixed and *D_g_* varies (Cases 1 and 2), and when *D_g_* is fixed and *D_gb_* varies (Cases 3, 4). Apparently, *D_g_* is more important in the mass transport process than *D_gb_*, whereas the grain occupies much more volume than GB. In most works only Cases 1 and 2 are analyzed, and only *D_g_* changes. We also wanted to examine the effect of *D_gb_* variation. Adsorption and desorption coefficients (when this process is included) are taken as α = β = 0.5. These values are quite realistic for a wide class of materials, and are similar to values we previously used for fitting experimental curves [35].

In Figure 3 five two-dimensional concentration depth profiles (concentration contours) are presented that are calculated using the different geometries of models (a), (b), (c), (d), and (e) (see Figure 2). A more intense yellow color means a higher concentration of diffusing atoms. Concentration contours are drawn according to a certain concentration value:(5)$$\mathrm{Case}\;1:\;c=\;\frac{0.6^{n}}{2}$$
(6)$$\mathrm{Case}\;2\;\mathrm{and}\;\mathrm{Case}\;3:\;c=\;\frac{0.1^{n}}{2}$$
where *n* is the value of concentration contour from left.

It is seen in Figure 3 that the concentration and penetration of diffusing atoms are much higher in Case 1. In all models, the concentration contours are curved starting from the surface. However, the concentration near the surface is distributed more evenly than deeper in the volume. It is seen that in Case 1, when *D_gb_/D_g_* = 10^3^, the curvature of the first contour is nearly invisible; in deeper layers, the curvature increases, but insufficiently. In Case 2, when *D_gb_/D_g_* = 10^4^, the curvature of concentration contours is well expressed and exhibits quite interesting geometry; moreover, it shows a steeper concentration gradient between grain and grain boundaries. The concentration contours in Case 1 correspond to the Harrison A regime [7] when the diffusion length is larger than the spacing between GB, so GB diffusion overlaps, forming less distorted concentration contours. Case 2 corresponds to the Harrison B regime, where GBs can be presumed to be isolated from each other. Atom transition from one GB to the second GB is insignificant, so the gradient between GB and grain is larger. According to [12,14], if more GB takes place (e.g., if the grains are small), more mass will be transferred into volume and the concentration gradient between grain and grain boundary will be steeper. In Cases 1 and 2, *Dg* differs, but *Dgb* is the same. To show the influence of *Dgb*, the Case 3 concentration contours are shown. The influence of *Dgb* is more pronounced for models with a smaller grain size, with a relatively higher volume of grain boundary *Vgb*. If we compare the concentration contours (Figure 2) in Cases 2 and 3 for large grain size models (e) and (d), it is seen that the concentration contours are almost the same. They differ for small grain size models (c), (b), and (a), i.e., when the relative volume of the grain boundaries increases.

In monocrystalline materials, diffusion is damped faster and the diffusion length is much smaller than in polycrystalline materials [5]. Furthermore, if a polycrystalline material is fine-grained, the effective diffusivity is larger than in coarser grains [22,36]. So, our results are in agreement with these statements. As the diffusion coefficients of grain and GB became more similar, the concentration gradient between grain and GB became smaller. It can be stated that Le Claire parameter β [1] is higher in Case 2, which means there will be a greater leakage from grain boundaries to grains.

From Figure 3 it can be seen that for Case 1 perpendicularity to surface GB influences the shape of concentration contours more efficiently than parallel ones. On the contrary, in Case 2, the concentration contours distort after meeting the parallel grain boundary. This can be seen more clearly in Figure 3 for Case 2 from the middle to the end of the calculated area. The existence of GB deeper in bulk has a significant influence on diffusion inside the grains. This occurs because atoms are virtually immobile in grains, whereas in GB they travel much faster; therefore, atoms that move in perpendicular GB can bend to parallel ones. Consequently, the concentration gradient between the grain and grain boundaries is larger when the *D_gb_/D_g_* coefficients ratio is higher.

The concentration profiles parallel to the surface at the middle of the investigated depth of samples *i = 140* are shown in Figure 4. Profiles are oscillatory, where minimums correspond with concentrations in the center of grains and maximums correspond with concentrations in the center of grain boundaries orientated perpendicularly to the surface of the sample. The curves of Case 1 and Case 2 show the influence of diffusivity in grains because *D_g_* was changed by an order of magnitude while *D_gb_* was constant. Curve 6 in Figure 4 is the calculated profile in samples without GB, which shows the influence of GB diffusion. In Case 1, the oscillation amplitude is not as large as in Case 2 because of the smaller relative diffusion coefficient *D_gb_/D_g_*. In Case 2 this ratio is higher, and the influence of grains is larger, which is reflected in the more prominent amplitude of the oscillations. In Case 1, the overall diffusivity is higher, because of the higher value of *D_g_* (see Table 1), and the curves are more separated than in Case 2. In Case 2, the positions of concentration oscillation minimums for different models are located very close to each other, but the maximums differ significantly. Likewise, according to [14], the smaller the grain diffusion coefficient *D_g_* the fewer atoms can diffuse from GB to grains; therefore, the difference between the maximum concentration in the center of the GB and the minimum one in the center of the grains becomes bigger. Otherwise, in Case 1 (Figure 4), the minimum points for different models are well separated. In Case 1, the amplitude between the concentration minimum and maximum points is highest in model (e), and lowest for model (a). Contrarily, in Case 2, the amplitude is highest in model (a), and lowest for model (e). This happens because the relative diffusion coefficient is lower than in Case 2, so grain boundaries affect the overall diffusion process, where the concentration gradient is smoother. These results correspond with research done by Han et al. [16]. They performed an analysis of different ratios of *D_gb_/D_g_*. They, like Bedu-Amissah [19], proclaimed a fixed grain diffusivity. The results were similar, with higher grain boundary diffusivity atoms traveling faster along grain boundaries so that an increase in overall diffusivity was observed; also, it boosts the concentration gradient between grain and grain boundaries.

The relative diffusion coefficient *D_gb_/D_g_* is not a very good parameter because it does not show the individual influence of GB and grains, and the results can significantly differ at the same value of ratio *D_gb_/D_g_*. In order to analyze that problem, the two other cases were considered: here they are named Case 3 and Case 4. The concentration profiles (calculated for model (b) in Figure 2) parallel to the surface at *i = 140*, calculated for all four cases, are shown in Figure 5 (the width is from *j = 90* to *j = 138*). Two figures are drawn in order to show the influence of the adsorption/desorption processes taking place on the surface. Results with and without adsorption/desorption are presented on the right and left side of Figure 5, respectively (indicated on the top of figure). Without adsorption/desorption means that the first layer always has a constant concentration equal to 1 a.u. In Cases 1 and 3, the *D_gb_/D_g_* ratio is the same; likewise in Cases 2 and 4, but the individual values of *D_gb_* and *D_g_* differ (see Table 2). In both profiles it is seen that Cases 2 and 3 and Cases 1 and 4 do not differ much, even though their *D_gb_/D_g_* ratio is different. Moreover, higher diffusion is seen where the relative diffusion coefficient is smaller. It follows that it is not the *D_gb_/D_g_* ratio that determines distribution of concentration, but the absolute values of the diffusion coefficients *D_gb_* and *D_g_*. The results correlate with an investigation done on diffusant uptake curves with different ratios of grain boundary and lattice diffusion coefficient (*D_gb_/D_l_*—relative boundary diffusion coefficient) by Bedu-Amissah and his research group [19]. They admitted that when the relative boundary diffusion coefficient is higher (the lattice diffusion coefficient is fixed, so only the grain boundary diffusion coefficient changes), more rapid filling of atoms occurs in the material. Comparing the curves with and without adsorption/desorption, it is seen that the curves of Cases 1 and 4 and Cases 2 and 3 almost correspond (because *V_gb_ << V_g_* (see Table 1) and the influence of GB is low when adsorption/desorption is not included. When adsorption/desorption is included, those curves differ even when they are calculated in a very deep layer (*i = 140*).

More details about the influence of adsorption/desorption are shown in Figure 6 for Cases 1 and 2, where the concentration profiles are drawn at the surface and nearby. When adsorption/desorption is neglected (Figure 6, left), the first layer profile is a straight line parallel to the surface because of the diffusion from the constant source. In deeper layers, a concentration redistribution takes place whereby the concentration is higher in GB because of the higher GB diffusion coefficient. Interesting profiles are seen in Figure 6 (right), where adsorption/desorption is included. In that case, the first layer profile is not a straight line and a decrease in GB concentration is observed. This can be explained by different diffusion coefficients of GB and grains, because in calculations, the adsorption and desorption coefficients for the grain and GB were taken the same. Due to the higher diffusion coefficient in GB atoms, they are more likely to travel faster, so they do not accumulate on the surface. It is interesting to point out that, in this case, atoms diffuse from grains to GB, because the concentration in grains becomes higher than in GB. That never occurs when adsorption/desorption is not included. In deeper bulk layers where the concentration becomes higher in GB than in grains, atoms diffuse from GB to grains in both cases, with and without adsorption/desorption. However, this phenomenon is not the most important influence on the adsorption/desorption process. Comparing profiles in deeper layers, it is seen that, in the case with adsorption/desorption (Figure 6, right), profiles are smoother than those without adsorption/desorption (Figure 6, left). This is more evident in Case 1 at a lower ratio of *D_gb_/D_g_*.

For better understanding how GB diffusion affects the overall diffusion process, a quantitative analysis was done. The concentration in grains of polycrystalline material only (concentration in grain boundaries is not taken into account) was compared with corresponding concentration after diffusion in monocrystalline material of the same volume using the same values of *D_g_*. This was done for all the different models in Figure 2. The formula used to express percentage difference is as follows:(7)$$\Delta c_{g\left(model\right)}=\frac{c_{g\left(model\right)}-c_{mono}}{c_{g\left(model\right)}}\times 100\%$$
where $c_{g\left(model\right)}$ is the concentration in grains for the corresponding model in Figure 2 and $c_{mono}$ is the concentration in a monocrystalline sample of the same volume. In Figure 7 the percentage changes over time are shown for Cases 1 and 2 and for each model. The time was taken as normalized $t_{1}<t_{2}<t_{3}$, and the time change interval between each time point is the same. Received data state that the smaller the grains, the bigger the concentration difference of diffusing atoms in polycrystalline materials compared with monocrystalline ones, assuming a greater diffusion coefficient in GB than in grains. In that case, more mass is transferred over GB than through grains, and around the grains the concentration is enlarged. More atoms are penetrating into the grain, which is why, further in, the crystal concentration gradient is larger compared to the first layers of the polycrystal [14,19,23,37].

In the above presented results, five models were analyzed. This many models were chosen in order to determine the functional dependence of the total number of diffused atoms on the volume of grain boundaries and crystal size. The parameter characterizing the volume of GB and crystal size is the ratio between the volume of grains and the volume of GB *V_g_/V_gb_*. Considering several models, it is possible to calculate the total amount of diffused atoms in the whole volume as a function of the ratio *V_g_/V_gb_*. One model provides one point of dependence. Our five models thus give five points of dependence that are necessary for interpolation, extrapolation, and analytical function evaluation. The calculated dependencies of average concentration in the whole volume on ratio *V_g_/V_gb_* are presented in Figure 8 (points). These results are presented at three different moments in time, for Cases 1 and 2, including or not including adsorption/desorption on the surface (Figure 8a,b (points), respectively). Those dependencies were plotted in order to obtain the analytical formula for the average concentration of ratio *V_g_/V_gb_*. Such a formula can be assumed to be the analytical solution of the presented model equations, Equations (2) and (4), which, because of the difficult boundary conditions, cannot be solved by simple integration methods. In order to obtain the formula, it is necessary to fit the calculated points with a certain function. However, the question remains of which function to choose for the fitting. It is well known that in the case of simple diffusion (no GB) from a constant source, the solution of Fick’s second law is error function *erf(x)*. In our case the situation is much more complex; nevertheless, the diffusion is described by Fick’s second law, only with many different boundary conditions and different diffusion coefficients. However, the function in form of error function *erf(x)* can be expected. So, the points in Figure 8a,b were fitted with the *erfc(x)* function. The fitting curves are presented in Figure 8 (lines), and go through all the points (accuracy > 99%). Very good fitting is obtained for all cases (Case 1, Case 2), including or not including adsorption/desorption processes. It can be seen in Figure 8a,b that the adsorption/desorption process lowers the total concentration of diffused material, but does not influence the functional dependence. The obtained formula is analytically written as:(8)$$c=a\times erfc\left(-\frac{b}{V_{g}/V_{gb}}\right)$$
where *a* and *b* are fitting parameters whose physical meaning needs to be found. The values of those coefficients are presented in Table 3. The obtained formula is very important because it allows us to evaluate the influence of grain boundaries’ volume and the size of grains on the diffusion process. Both coefficients depend on time. In order to better understand the physical meaning of those coefficients, their time dependencies are plotted in Figure 9. The time dependencies of coefficient *a* for different cases are plotted in Figure 8a (points). A monotonic increase with time is observed in all cases. Those points were fitted using square root function *t^1/2^*. The lines in Figure 9a are the fitting results. For all cases, the fit is very good. So, the function of time of coefficient *a* is as follows: $a\left(t\right)=q\sqrt{t}$, where the *q* values for each fitting are listed in the legend of Figure 9a. The time dependence of coefficient *b* for different cases is plotted in Figure 8b. The dependence on time of coefficient *b* is not clear. At the beginning, *b* increases with time and then slowly decreases for Case 1. For Case 2 it is the opposite: it decreases at the beginning and then quite speedily increases. Dependence of coefficient *b* on diffusion coefficients *D_gb_* and *D_g_* can be expected. Processes of adsorption/desorption slightly influence these values, but the tendency remains the same. No general function for fitting can be proposed, and the lines in Figure 9b are just point connections.

## 4. Conclusions

(1)Adsorption and desorption processes taking place on the surface have a significant influence on the distribution of diffusing atoms and can qualitatively change the concentration profile curves parallel to the surface.(2)Not the relative diffusion coefficient *D_gb_/D_g_* but the absolute values of both diffusion coefficients (grain boundary, *D_gb_* and grain, *D_g_*) determine the concentration distribution.(3)The shape of concentration profile curves parallel to the surface becomes more distorted when the relative diffusion coefficient *D_gb_/D_g_* increases.(4)The average concentration of diffused atoms over the whole volume depends on the ratio *V_g_/V_gb_* according to the complementary error function.

## Figures and Tables

**Figure 1 materials-13-01051-f001:**
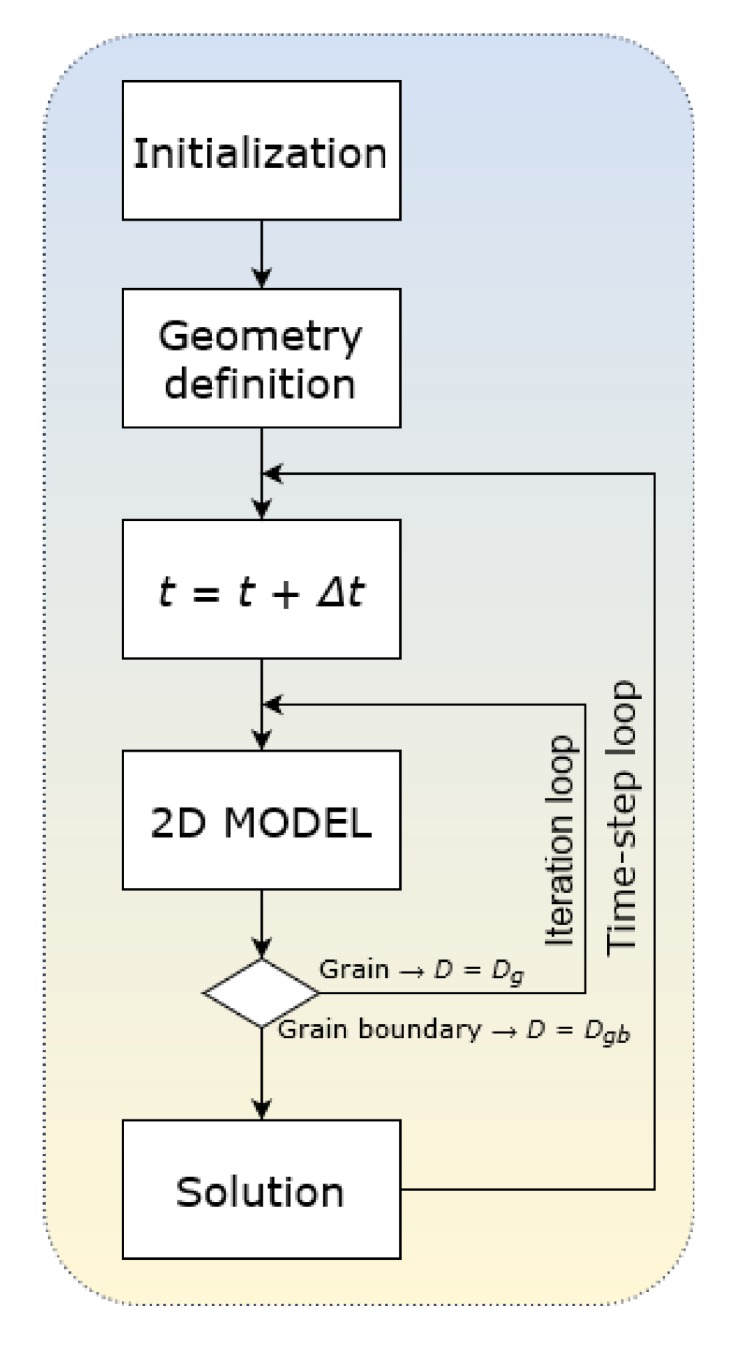
Schematic presentation of the solver.

**Figure 2 materials-13-01051-f002:**
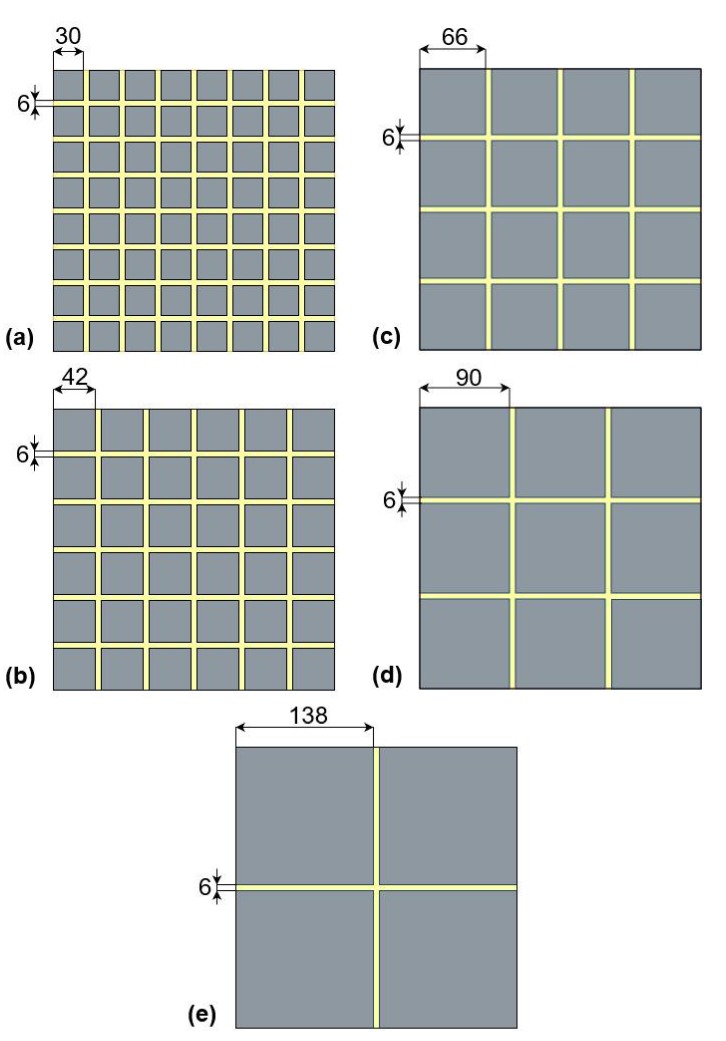
Geometry of models with different grain sizes, in a.u.: (**a**) 30, (**b**) 42, (**c**) 66, (**d**) 90, and (**e**) 138. Grain boundary width is fixed for all models at 6 a.u.

**Figure 3 materials-13-01051-f003:**
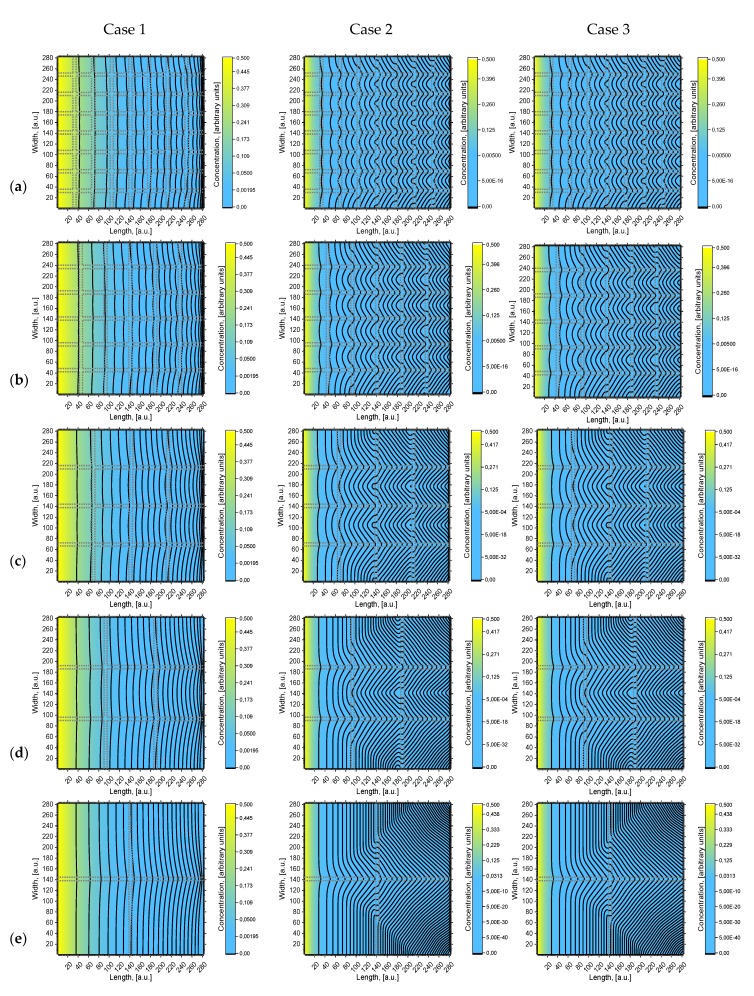
Calculated concentration distribution images with concentration contours of models (**a**–**e**) (see Figure 2) for Case 1, Case 2, and Case 3 (see Table 2).

**Figure 4 materials-13-01051-f004:**
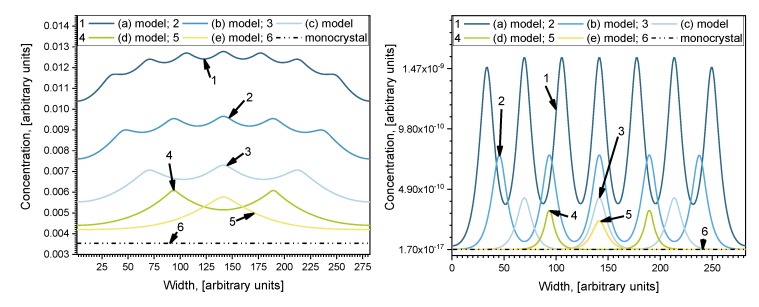
Parallel to the surface concentration profiles for different models at position *i* = 141 a.u.: at left side for Case 1 and at right side for Case 2.

**Figure 5 materials-13-01051-f005:**
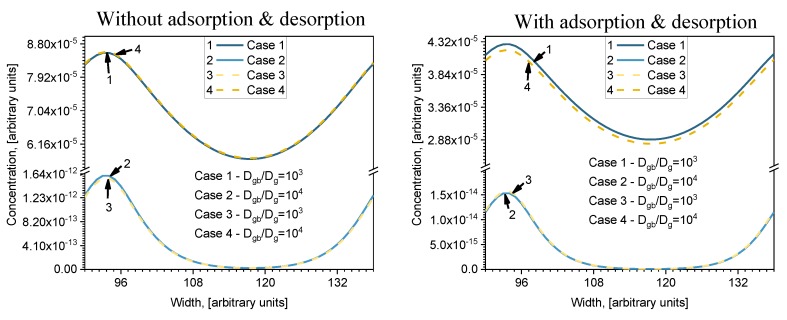
Parallel to the surface concentration profiles for (**b**) model (see Figure 2) at depth *i* = 141 a.u: at left side without adsorption/desorption and at right side with adsorption and desorption.

**Figure 6 materials-13-01051-f006:**
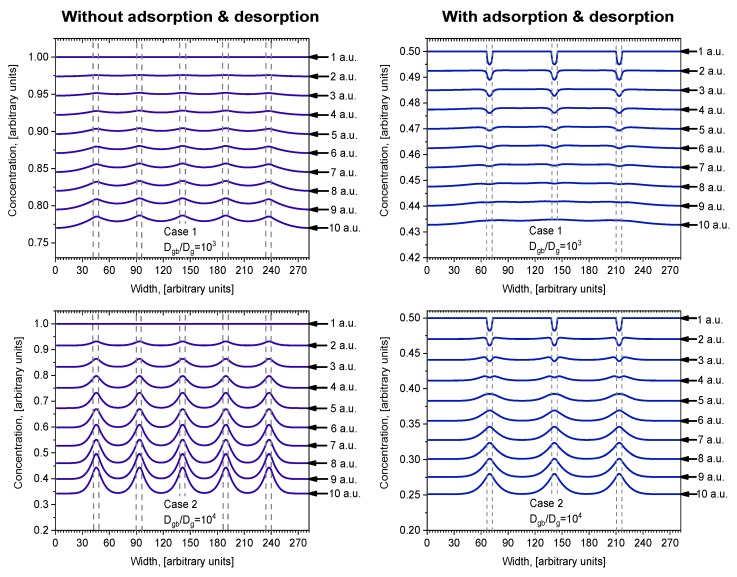
Parallel to the surface concentration profiles for the first 10 monolayers: at left side without adsorption/desorption and at right side with adsorption and desorption; at top for Case 1 and at bottom for Case 2.

**Figure 7 materials-13-01051-f007:**
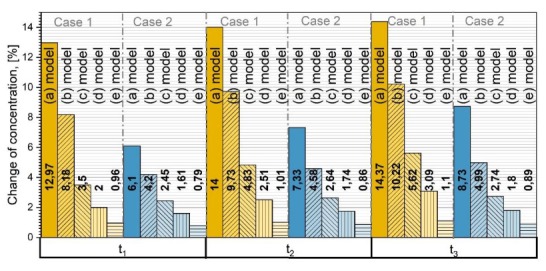
The quantitative comparison of diffusion in grains of polycrystalline material of different models (see Figure 2) with corresponding monocrystalline material of the same volume at three different moments of time and for Cases 1 and 2.

**Figure 8 materials-13-01051-f008:**
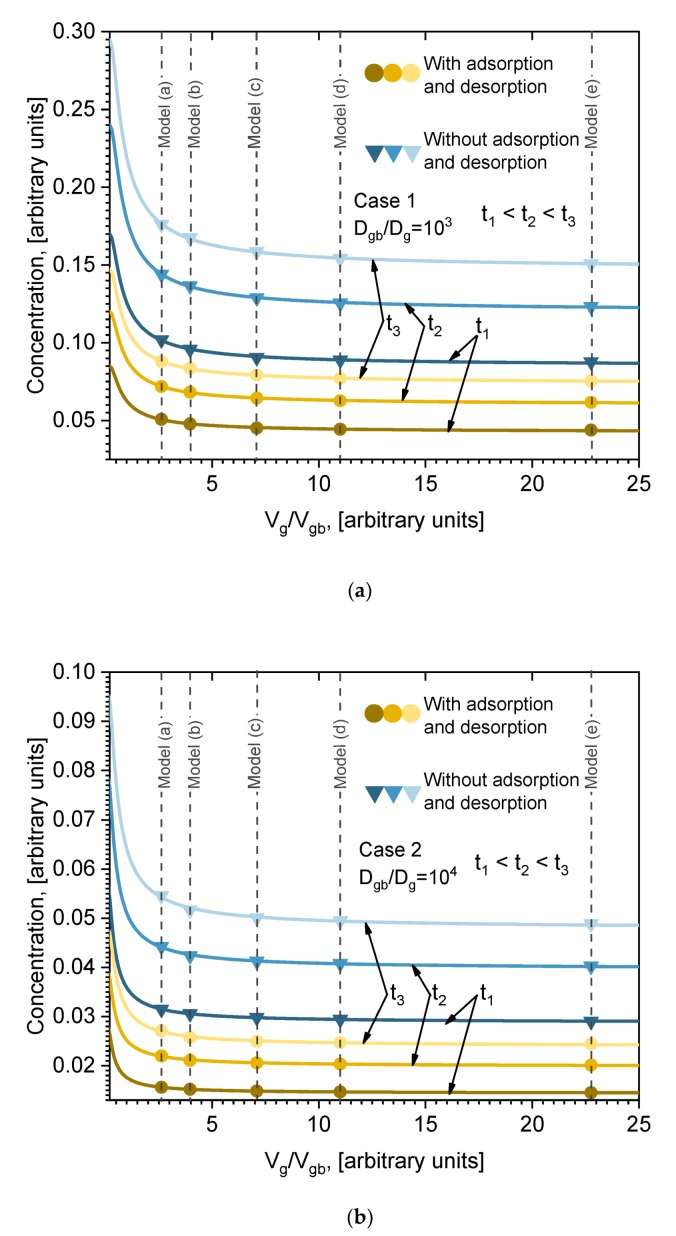
Average concentration dependencies (points) on relative volume *V_g_/V_gb_* at different moments of time and for cases with and without adsorption/desorption. Lines are fitting results using the error function in Equation (8): (**a**) Case 1; (**b**) Case 2.

**Figure 9 materials-13-01051-f009:**
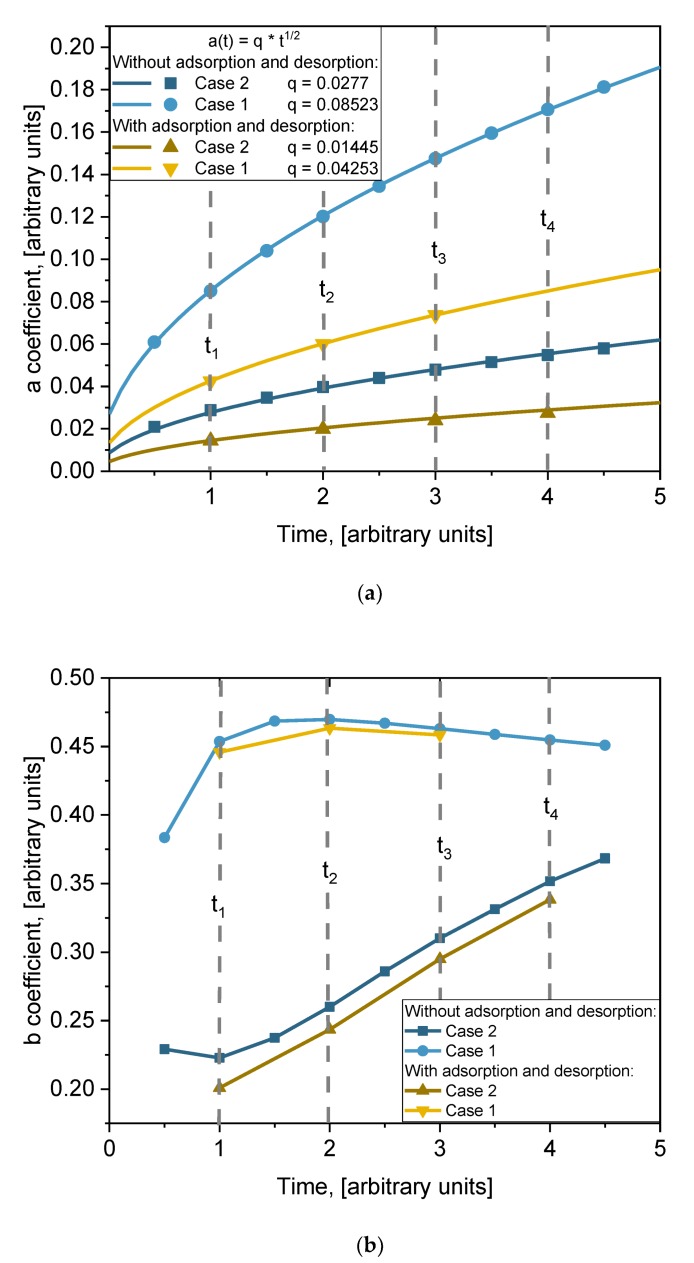
Dependencies on time (points) of coefficients (**a**) *a* and (**b**) *b* from Equation (8) (values from (Table 3)) for Case 1 and Case 2 with and without adsorption/desorption processes. In (**a**) lines are fitting of points with function *q**⋅t^1/2^*, and q values are indicated in (**a**).

**Table 1 materials-13-01051-t001:** Comparison of two-dimensional volume for each model. *V*—two-dimensional volume of the test material, *V_g_*—grain two-dimensional volume, *V_gb_*—grain boundary two-dimensional volume.

Volume	(a) Model	(b) Model	(c) Model	(d) Model	(e) Model
V (a.u.^2^)	79,524	79,524	79,524	79,524	79,524
V_g_ (a.u.^2^)	57,600	63,504	69,696	72,900	76,176
V_gb_ (a.u.^2^)	21,924	16,020	9828	6624	3348
Percentage of grain boundaries according to whole two-dimensional volume (%)	28%	20%	12%	8%	4%

**Table 2 materials-13-01051-t002:** Values of diffusion coefficients in different cases. *D_gb_*—grain boundary diffusion coefficient, *D_g_*—grain diffusion coefficient, *D_gb_/D_g_*—relative diffusion coefficient.

Cases	*D_gb_*	*D_g_*	*D_gb_/D_g_*
Case 1	0.9	0.0009	10^3^
Case 2	0.9	0.00009	10^4^
Case 3	0.09	0.00009	10^3^
Case 4	9	0.0009	10^4^

**Table 3 materials-13-01051-t003:** Discovered values of *a* and *b* coefficients in both Cases 1 and 2 (with and without adsorption/desorption).

Time	With Adsorption and Desorption				Without Adsorption and Desorption			
	Case 1		Case 2		Case 1		Case 2	
t [a.u]	a	b	a	b	a	b	a	b
0.5	-	-	-	-	0.06089	0.38351	0.02091	0.2292
1	0.04251	0.4458	0.01437	0.2009	0.08505	0.45366	0.02875	0.2228
1.5	-	-	-	-	0.10398	0.46853	0.03471	0.2374
2	0.06006	0.4634	0.01983	0.2435	0.12013	0.46974	0.03967	0.2601
2.5	-	-	-	-	0.13446	0.46696	0.044	0.286
3	0.07373	0.4584	0.02394	0.295	0.14748	0.46299	0.04789	0.3102
3.5	-	-	-	-	0.15949	0.4588	0.05146	0.3314
4	-	-	0.02739	0.3384	0.17069	0.45474	0.05478	0.3518
4.5	-	-	-	-	0.18122	0.45093	0.05791	0.3684

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
