# Peer review of "Kinetic Modeling of Grain Boundary Diffusion: The Influence of Grain Size and Surface Processes"

_materials, 2020, doi:10.3390/ma13051051_

Round 1

Reviewer 1 Report

Manuscript is well written and results are clearly presented, but the manuscript must be revised; authors should describe the aims and the potential applications of their research. What is the purpose of the research?

Author Response

Polycrystalline materials are composed of a large number of grains, therefore grain boundaries can work as a sink or as an accelerating path. Consequently, there is a possibility of an increase or reduction of diffusion. This can be applicable in the field of ceramics and in hydrogen fuel cells. The purpose of the research is to have a tool (model and code) to consider mechanisms of diffusion in polycrystalline materials where many aspects are not fully understood.

Text is added

The purpose of this research is to have a tool (model and code) to consider dynamics and mechanisms of diffusion of atoms and ions in polycrystalline materials which many aspects are not fully understood, especially ions in superionic materials

Reviewer 2 Report

I have read the manuscript, and the only remark I have to the authors is to write a sentence about potential applications of the research. Also, improve the quality of the figures 2, 3, 4, 7 and 8.

Author Response

Originally quality of figures is quite good, the quality may decrease when transfer files through the publisher online system.

Text is added

In this work we want to obtain a better understanding of how atoms are transferred through different types of geometry of nanocrystalline materials, to ascertain how grain boundaries and their occupied area influence concentration change in grains. Also, to show what the influence of processes on the surface is, especially adsorption and desorption, because, most of the models ignore this question and use constant source in the first layer. The purpose of this research is to have a tool (model and code) to consider dynamics and mechanisms of diffusion of atoms and ions in polycrystalline materials which many aspects are not fully understood, especially ions in superionic materials.

Reviewer 3 Report

The paper describes the model of diffusion in polycrystalline material using two dimensional diffusion equation. Two dimensional rectangular geometry with two opposite edges acting as asorbtion/desorbtion and desorbtion source is used. One edge acts as adsorbtion/desrbtion surface and is  contacte by media of 1 a.u. (arbitrary units) concentration. One edge is desorbtion edge in contact with 0 a.u. medium . The adsorbtion/desorbtion is modelled by Langmuir equation. An explicit numerical scheme based on the first order central difference discretization is used to solve the equation. The average concentration of diffused atoms c is related to the volumetric ratio of grain boundaries to grain interiors (Vg/Vgb) as c=a(t)*erfc(b(t)/(Vg/Vgb)); The function a(t) is suggested in form a(t)=q*sqrt(t). Function b is not fitted. The article represents interesting modelling data leading to empirical relation c on grain size characterized by Vg/Vgb. However to fully exploit its potential, it must be presented more rigorously and all the relevant detail should be given. Therefore I recommend Major revision.

Major comments:
The empirical formula (8) should be discussed with other available models or even better with some experimental results

The solver is not described in sufficient detail. It is not indicated whether the results represent concentration at arbitrary time or steady or quasi steady-state.

Parameters used for Langmuir equation are not given. What is their value, how do they correspond to the chosen values od diffusion coefficients. Can any material nave such values?

Boundary conditions for concentrations are not given explicitly. What is the boundary condition on the edges of the region of interest that are in contact with other material? (up and down in Fig. 2).

Isoclines are mentioned in Fig 2 and elsewhere, these seem to be no isoclines (lines of constant time derivative of concentration), but rather concentration contours.

The discussion on the Dgb/Dg states that both Dgb and Dg are necessary, but the results show tha Dg is defining the concentration, regardless of Dgd at least without adsobtion. Moreover the effect of Dgb in increased concentrations observed in [19] (line 227) is not observed in (figure 4).

If would be reasonable to also include cases and 3 in a similar fashion as figure 2 as Dgb changes there.

I is not clear how were the values of diffusion constants selected, do they correspond to some values found in specific materials?

How is cg defined ? as an average grain concentration of all grains ?

Minor comments:

Model number missing in figure 4.

References to figures are broken.

Author Response

1. Formula (8) is firstly (in our knowledge) postulated by us, which follows from our calculations. At the moment it is difficult to it with available models and experimental results as nobody calculated or measured experimentally (is it possible?) concentration dependence on G and GB volume ratio. However, results presented here and from which we obtain formula (8) are discussed with model and with results, so formula (8) must be in agreement.

But it is an interesting question and in future we will work with that formula looking for theoretical and experimental agreement, here is the first attempt to publish it.

2. Results present concentration at a taken arbitrary time.

Text is added

All the calculated results presented below are outputted at a certain arbitrary time.

3. 

Parameters of Langmuir equation are:  adsorption coefficient a= b=0.5. They are quite realistic similar we used previously for fitting experimental points

Text is added

Adsorption and desorption coefficients when this process is included are taken as α=ß=0.5. These values are quite realistic for a wide class of materials, similar values we previously used for fitting experimental curves [35]

4.

In fig.2: the surface is left edge where adsorption/desorption takes place then adsorbed atoms penetrate into the material and diffuses according 2d geometry. When approaches right edge atoms can desorb and leave the material. At lateral edges (in fig 2 up and down) the desorption is excluded, atoms cannot leave material from lateral edges. 

Text is added

Boundary conditions are taken following: atoms adsorb onto the surface, part of them can desorb, another part penetrates surface layer and diffuses in the volume of material according to 2-d geometry. Those atoms which reach the bottom edge layer may desorb. Desorption from lateral edge layers is excluded.

5

We think that “concentration isoclines” is synonymous to “concentration contours”, we suppose not to change

6

Not only Dgb has influence but Vgb also (volume of grain boundaries). If volume Vgb is low comparing with Vg the influence Dgb is also low. This occurred in the case considered in fig. 4. Values of Vg and Vgb are given in Table 1, Vgb for b model is 20% of the whole volume, what why the effect of Dgb is not significant. But in general effect that concentration increases with increase of value of Dgb is seen in fig 4 (without adsorption/desorption), as curve 4 (Dgb=9) is above curve1 (Dgb=0.9) and curve 2 (Dgb=0.9) is above curve 3 (Dgb=0.09). So results are in agreement with [19].

The comment in the text of the manuscript is added:

(because Vgb<<Vg (see table 1) and influence of GB is low)

7

In Fig 2 Case 3 is added

Text is added

In Case 1 and Case 2 differs Dg, and Dgb is the same. To show the influence of Dgb the Case3 isoclines are shown. The influence of Dgb is more observed for smaller grains size models, with a relatively higher volume of grain boundary Vgb. If to compare isoclines of Fig.2 Case 2 and Case 3 for big grain size models (e) and (d) it is seen that isoclines are almost the same. They become to differ for small grain size models (c), (b), (a), i.e. when the relative volume of grain boundaries increases.

8.

The values of diffusion constants are in Table 1. Absolute values are in arbitrary units, so they always can be adapted for real materials. Relative diffusion constant was selected according to properties of polycrystalline materials where the ratio is usually in interval Dgb/Dg ~ 102-106. Interval 102-104 is considered in [19].

Text is added

Ratio Dgb/Dg in the interval 102-104 is most usually taken for consideration [19].

9. cg is average concentration in all grains.

10. 

Model number is added, (b) model

References to figures were broken the first version of pdf file 

Reviewer 4 Report

In their contribution ”Kinetic Modeling of Grain Boundary Diffusion: Influence of Grain Size and Surface Processes”, Jaseliunaite and Galdikas report on the development of a code to model diffusion processes at ground boundaries. In addition, the authors present the application of this code to diverse models. Although the contribution appears to be suited for materials, yet, there are certain issues, which should be solved prior to a publication of the manuscript:

- Although a brief description regarding the kinetic model that has been employed in the present study has been provided, there is no detailed description about the work flow of the solver that was designed using C++. Therefore, it will be helpful to provide a chart describing the work flow of the program. Furthermore, the authors did not provide any information regarding the resources, which were needed to compute their models. Additionally, the authors should state, if this small code is made available to the public.

- In the introduction, it is mention that Monte Carlo simulation have been employed in previous studies. Because Monte Carlo simulation can be quite costly in terms of time and resources, it will be interesting to see, if the procedure shown in the present contribution allows minimizing the required computational time and resources. Hence, the authors should include such comparison between their proposed procedure and alternative strategies.

- It is hard to understand what the authors mean by “30 a.u. model (a), 42 a.u. model …”. Hence, the authors should provide more information about the starting models used in the computations.

- There are certain typos, e.g. “… with whose statements” (line 172), that should be fixed.

Author Response

There is given equations in 2 part of a manuscript by which program was developed. The program is based on the finite differences method and explicit discretization scheme. Initial conditions and parameters are given.

The code is still under development in order to provide more functions. The code is not public.

Text is added

. The schematic presentation of the solver is shown in Fig.1. Boundary conditions are taken following: atoms adsorb onto the surface, part of them can desorb, another part penetrates the surface layer and diffuse in the volume of material according to 2-d geometry. Those atoms which reach the bottom layer may desorb. Desorption from lateral edge layers is excluded.

The Figure 1 Schematic presentation of the solver is added.

Round 2

Reviewer 2 Report

The manuscript is now suitable for publication in Materials.

Author Response

Thank You

Authors

Reviewer 3 Report

The authors adressed most of reviewers comments and the articles is now much easier to follow. There are still two of the original questions that should be adressed in more details:

Q1 Boundary conditions for concentrations are not given explicitly. What is the boundary condition on the edges of the region of interest that are in contact with other material? (up and down in Fig. 2).

The authors provided the b.c., but did not connect it with any real situation, i.e. did not described what is modelled in reality. In further works I suggest a periodic b.c. at lateral edges. It will get closer to simple 2d infinite plate geometry, and reduce computational complexity as only one column of grains has to be analyzed. A 3D model with periodic BC would be even closer to real problem.

Isoclines are mentioned in Fig 2 and elsewhere, these seem to be no isoclines (lines of constant time derivative of concentration), but rather concentration contours.

The authors are incorrect: Isoclines are lines with same slope (gradient). The article contains contours or isolines

Author Response

The real situation is e.g. in hydrogen fuel cell

Text is added describing boundary condition

Such boundary conditions let us simulate mass transfer processes in polycrystalline electrolytes of solid oxide hydrogen fuel cells (SOFC), where oxygen ions diffuse from one cathode to anode which is placed on another side of electrolyte layer and then desorb after recombination with hydrogen. Removal (and arrival) of particles from (to) the lateral surfaces is excluded or negligible.

Text is added at the end of Introduction

The boundary conditions of the proposed model correspond with the real situation in electrolytes of solid oxide hydrogen fuel cells (SOFC) and can be applied for simulation of mass transport processes in polycrystalline electrolytes.

Text is added in Abstract

The boundary conditions of the proposed model correspond with the real situation in the electrolytes of solid oxide hydrogen fuel cells (SOFC).

Added to keywords

Solid Oxide Fuel Cells